# Silent Antibodies Start Talking: Enhanced Lateral Flow Serodiagnosis with Two-Stage Incorporation of Labels into Immune Complexes

**DOI:** 10.3390/bios12070434

**Published:** 2022-06-21

**Authors:** Dmitriy V. Sotnikov, Nadezhda A. Byzova, Anatoly V. Zherdev, Youchun Xu, Boris B. Dzantiev

**Affiliations:** 1A.N. Bach Institute of Biochemistry, Research Center of Biotechnology of the Russian Academy of Sciences, Leninsky Prospect 33, 119071 Moscow, Russia; nbyzova@mail.ru (N.A.B.); zherdev@inbi.ras.ru (A.V.Z.); dzantiev@inbi.ras.ru (B.B.D.); 2State Key Laboratory of Membrane Biology, Department of Biomedical Engineering, School of Medicine, Tsinghua University, Beijing 100084, China; xyc2012@mail.tsinghua.edu.cn

**Keywords:** immunochromatography, test strips, RBD protein, COVID-19, coronavirus

## Abstract

The presence of pathogen-specific antibodies in the blood is widely controlled by a serodiagnostic technique based on the lateral flow immunoassay (LFIA). However, its common one-stage format with an antigen immobilized in the binding zone of a test strip and a nanodispersed label conjugated with immunoglobulin-binding proteins is associated with risks of very low analytical signals. In this study, the first stage of the immunochromatographic serodiagnosis was carried out in its traditional format using a conjugate of gold nanoparticles with staphylococcal immunoglobulin-binding protein A and an antigen immobilized on a working membrane. At the second stage, a labeled immunoglobulin-binding protein was added, which enhanced the coloration of the bound immune complexes. The use of two separated steps, binding of specific antibodies, and further coloration of the formed complexes, allowed for a significant reduction of the influence of non-specific immunoglobulins on the assay results. The proposed approach was applied for the serodiagnosis using a recombinant RBD protein of SARS-CoV-2. As a result, an increase in the intensity of test zone coloration by more than two orders of magnitude was demonstrated, which enabled the significant reduction of false-negative results. The diagnostic sensitivity of the LFIA was 62.5% for the common format and 100% for the enhanced format. Moreover, the diagnostic specificity of both variants was 100%.

## 1. Introduction

Detection of pathogen-specific antibodies in the blood (serodiagnosis) plays a key role in the diagnosis of many infectious diseases. The advantages of this method are the selection of a specific biomatrix for testing (in contrast with the determination of viruses or bacteria in different organs and tissues), and the similarity of the assay protocols for various infections [1]. In medical diagnostics, microplate-based enzyme-linked immunosorbent assays (ELISAs), agglutination tests, and other immunoassay formats are successfully used for this purpose [2,3,4]. The current trend in the development of diagnostic tools is simplification and acceleration of the testing procedure. The lateral flow immunoassay (LFIA) is one of the most relevant immunoassay formats, with the availability of industrial facilities for the wide manufacturing of diagnostic kits. Multimembrane composites (test strips) with preliminary applied immune reactants and colored nanoparticles as detected labels provide maximum reducing manipulations of an operator. Contact of a test strip with the tested sample initiates the lateral flow of the immune reactants along the membranes and the assay results can be visualized within 10–15 min by the coloration of the test strip zones [5]. Successful development of the LFIA-based serodiagnosis of various diseases is presented in several publications [6,7,8,9,10]. As an additional application, the assessment of the vaccine effectiveness and the organism’s protection against infections can be mentioned [11,12].

The recent COVID-19 pandemic also caused the necessity of serodiagnostic LFIAs, which were successfully developed and commercialized by different groups and organizations [13,14]. Nevertheless, the development and practical application of a serodiagnostic LFIA are accomplished by significant limitations. The percentage of the infection cases revealed by the LFIA serodiagnosis is often inferior to the cases detected by instrumental immunoassays, such as the ELISA. The main factor causing this obstacle is the need to detect specific antibodies with low concentrations in the presence of great excess of total immunoglobulins [15].

Serodiagnosis is typically implemented using immunoglobulin-binding proteins, such as anti-species antibodies or bacterial protecting compounds (staphylococcal protein A, streptococcal protein G, etc.) [16]. In the common LFIA format, they are immobilized on colored nanoparticles, applied on a test strip, and form complexes with all immunoglobulins, including those specific to the target pathogen during lateral flow. The resulting complexes interact with the antigen immobilized in the test zone (TZ) of the strip, whereas unbound compounds move across this zone. Therefore, the coloration of the TZ indicates the presence of specific antibodies in the tested sample [8,17,18,19]. However, non-specific immunoglobulins block binding sites at the surface of nanoparticles and only a minor part of specific antibodies can form labeled complexes in the TZ. This differs from the ELISA-based serodiagnosis where non-bound antibodies can be removed (washed) prior to the inclusion of labels into immune complexes. This limited binding leads to a significant percentage of false-negative results, which are often indicated during the validation of these assays [20,21].

The possibility to change the order of the immune complexes formation was considered in alternative serodiagnostic LFIAs [20,22,23,24]. As a result, some limitations were noted in these approaches. For example, the inverted LFIA format, where the antigen is immobilized on label particles and the immunoglobulin-binding protein is adsorbed on the working membrane, allows for the binding of more immunoglobulins than the traditional LFIA format. Nevertheless, in this case, the interference of non-specific immunoglobulins is not eliminated [25,26,27]. One of the approaches to eliminate their influence is the “double antigen sandwich” LFIA format based on the antibody polyvalence [23,28,29,30]. However, due to the side formation of polyvalent complexes, the efficiency of this format strongly depends on the ratio of the reagents [15]. Moreover, the use of more sensitive labels or signal amplification by aggregation or catalysis was considered [31,32,33,34]. The improvement achieved by these methods is associated with significant changes in the production of test strips. Therefore, they can be implemented in mass practice only after solving additional tasks, such as stabilization of new reactants and prevention of their non-specific interactions.

Therefore, the improvement of the common serodiagnostic LFIA based on the conjugates of immunoglobulin-binding proteins with traditional nanodispersed labels that ensure an increase in labeled immune complexes in the TZ is in great demand. In the present study, a two-stage serodiagnostic LFIA is proposed. This method includes the common LFIA (the first stage) and additional labeling of bound specific immunoglobulins by a conjugate of gold nanoparticles with an immunoglobulin-binding protein (the second stage). This approach was performed for the serodiagnostic of COVID-19 infection and demonstrated the enhancement of the coloration intensity by two orders of magnitude, as compared with the common LFIA format, which significantly reduces the likelihood of false-negative results.

## 2. Materials and Methods

### 2.1. Chemicals, Materials, and Apparatuses

The recombinant receptor-binding domain (RBD) of SARS-CoV-2 spike protein [35] was kindly provided by Dr. I.I. Vorobiev. The monoclonal antibodies (MAb) to RBD, clone RBD5313, were from HyTest (Moscow, Russia). Goat anti-mouse immunoglobulins (GAMI) and a conjugate of recombinant staphylococcal protein A with gold nanoparticles (pA–GNP) were from Arista Biologicals (Allentown, PA, USA). The polyclonal anti-human antibodies labeled with horseradish peroxidase were from Imtek (Moscow, Russia). NIBSC Anti-SARS-CoV-2 Antibody Diagnostic Calibrant reagent was purchased from National Institute for Biological Standards and Control (Hertfordshire, UK). Human serum was kindly provided by Dr. S.F. Biketov (State Research Center of Applied Microbiology and Biotechnology, Obolensk, Russia) and collected during previous studies [33] from volunteers and patients after obtaining written and informed consent. The pooled negative serum was prepared by mixing 10 sera from donors without symptoms of respiratory diseases and antibodies against RBD of SARS-CoV-2, in accordance with enzyme immunoassay.

Bovine serum albumin (BSA), sucrose, poly(vinyl formal), Tris, Tween 20, Triton X-100, 3,3′,5,5′-tetramethylbenzidine (TMB), hydrogen peroxide (H_2_O_2_, 30%), were from Sigma-Aldrich (St. Louis, MO, USA). Salts and acids were all from Khimmed (Moscow, Russia). The solutions were all prepared using deionized water produced by Milli-Q (Billerica, MA, USA).

Nitrocellulose working membranes (CNPC-15), glass fiber conjugate membranes (PT-R7), sample membranes (GFB-R4), and adsorbent membranes (AP045) were purchased from Advanced Microdevices (Ambala Cantt, India). The 96-well transparent polystyrene microplates for ELISA were purchased from Corning Costar (Tewksbury, MA, USA).

### 2.2. Characterization of pA–GNP

For transmission electron microscopy (TEM), pA–GNP solution was applied to 300-mesh grids (Pelco International; Redding, CA, USA) coated with a poly(vinyl formal) film. Then, the film was placed on the glass and exposed to 0.15% *v*/*v* solution of formvar in chloroform. The images were obtained with a JEM CX-100 microscope (Jeol, Tokyo, Japan) at 80 kV and analyzed by the Image Tool software (University of Texas, Health Science Center, San Antonio, TX, USA).

The hydrodynamic size of pA–GNP was measured using a Zetasizer Nano (Malvern Pananlytical; Malvern, UK) at 25 °C for 10 s at a scattering angle of 173°.

### 2.3. Preparation of Test Strips

GAMI (0.5 mg/mL) and RBD (0.5–1.0 mg/mL) in 50 mM K-phosphate buffer with 100 mM NaCl, pH 7.4 (PBS), were applied to the control zone (CZ) and the TZ of the working membrane, respectively, by Image Technology IsoFlow dispenser (Lebanon, NH, USA) at a loading of 0.12 µL/mm. The pA–GNP conjugate (optical density at 520 nm (OD_520_) = 2–20) containing 1.0% *v*/*v* Tween 20 was applied to the conjugate membrane (0.8 µL/mm). Both membranes were dried at room temperature for 12 h and composed together with sample and absorbent membranes to multimembrane sheets. Finally, test strips (of 3.5 mm width) were obtained by cutting sheets with an automatic Index Cutter-1 guillotine (A-Point Technologies; Gibbstown, NJ, USA) and stored at room temperature in zipper bags.

### 2.4. Lateral Flow Immunoassay

The common LFIA was performed at room temperature as follows:−The test strip was placed on a horizontal surface;−60 µL of a sample was applied to the sample membrane;−the strip was incubated for 5 min;−20 µL of TTBSA buffer (10 mM Tris, 0.25% *w*/*v* BSA, 0.25% *w*/*v* sucrose, 1.0% *v*/*v* Tween 20, 0.05% *w*/*v* NaN_3_, pH 8.5) was applied to the sample membrane;−the strip was incubated for 5 min.

The enhanced LFIA was started as described above for the common LFIA. Then, 2 μL of the pA–GNP conjugate was applied to the bottom edge of the working membrane. Thereafter, 60 μL of TTBSA was applied and the test strip was incubated for 5 min.

Following the LFIA, test strips were scanned using a CanoScan 9000F scanner (Canon, Tochigi, Japan). The obtained digital images were processed by TotalLab TL120 software (Nonlinear Dynamics, Newcastle, UK) to measure the intensity of TZ coloration. The processing of test strips by a TotalLab application included the finding of TZ in the scanned images, generation of color intensity profiles, subtraction of background coloration using the “Background Subtraction” tool, and final registration of the “Volume” value as the analytical signal using the “Band Detection” tool.

Each sample was tested in duplicate. The limit of detection (LOD) was determined as the concentration providing the coloration of the TZ, which is higher than the sum of the average coloration intensity and three standard deviations for a blank probe.

### 2.5. ELISA of Human Serum

RBD (1 μg/mL, in PBS) was incubated in the microplate wells at 4 °C overnight. The wells were washed 4 times to remove unbound molecules using PBS with 0.05% *v*/*v* Triton X-100 (PBST). Thereafter, serum diluted with PBST (1:25–1:50,000) was added and the microplate was incubated for 1 h at 37 °C. Then, the microplate was washed again and anti-human antibodies labeled with horseradish peroxidase (dilution of 1:3000 in PBST) were added to each well. The microplate was incubated for 1 h at 37 °C. After washing the microplate as described above, the enzyme activity of the bound peroxidase label was determined. For this purpose, the substrate mixture containing 0.4 mM TMB and 3 mM H_2_O_2_ in 40 mM sodium citrate buffer, pH 4.0, was added. Following incubation at room temperature for 15 min, the reaction was stopped by adding 1 M H_2_SO_4_ to the substrate mixture (*v*/*v* = 1:2). Finally, A_450_ was measured using a Zenyth 3100 microplate photometer (Anthos Labtec Instruments, Wals, Austria).

## 3. Results and Discussion

### 3.1. Consideration of the Proposed Assay Format

The serodiagnostic LFIA can be implemented in several formats, which differ in the order of arranging the reagents on the membranes and the composition of the resulting complexes [20,22,23,24]. The most common format is based on the use of labeled immunoglobulin-binding proteins as reagents to visualize immune complexes. Pathogen antigens are immobilized in the TZ and interact with specific antibodies, which leads to label binding.

The assay proposed in this study was first implemented in the same manner as the common LFIA (Figure 1, stage 1). In the case of the presence of specific antibodies in a tested blood sample, colored complexes of the immobilized antigen, specific antibodies, and a labeled immunoglobulin-binding protein are formed in the TZ. To clarify points for improvement, let us consider this process in more detail. The concentration of IgG as the main class of immunoglobulins in human blood is typically about 6–20 mg/mL [36], while the sorption capacity of gold nanoparticles with a diameter of 30 nm (as the optimal LFIA label) does not exceed 5 μg/mL per unit of the OD [37]. Therefore, even if the OD of the conjugate is 10, it binds less than 1% of the IgG fraction from the sample. The same proportion is typical for specific anti-pathogen antibodies. Theoretically, the problem can be solved by concentrating the conjugate. However, an increase in the concentration of gold nanoparticles promotes their aggregation, which leads to overlapping of binding centers and a stability decrease. Therefore, a radical increase in the conjugate concentration is restricted. As a result, the majority of specific antibodies that bind in the TZ are not associated with the label. To involve them in the generation of the detected signal, an additional stage was proposed.

At the second stage of the assay, a labeled immunoglobulin-binding protein is added to the working membrane of the test strip. This complex flows along the membranes after the sample, thus washing out the unbound immunoglobulins from the working membrane and interacting with the immunoglobulins bound in the TZ. At this stage, the release of the analytical signal occurs in two ways (Figure 1, stage 2). On the one hand, the labeled immunoglobulin-binding protein interacts with the antibody-antigen complexes in the TZ. On the other hand, the label conjugate bound in the TZ at the first stage provides available sites to bind the added reagent. Since the IgG molecules have two symmetric binding sites for immunoglobulin-binding proteins, they could serve in the formation of various polycompound complexes based on the labeled immunoglobulin-binding protein, which is initially bound at the TZ and contains a large number of IgG molecules at its surface. In this way, non-specific immunoglobulins interfere with signal generation at the first stage, but provide signal enhancement at the second stage.

Aggregation of functionalized nanoparticles was previously used in LFIA to enhance the analytical signal [38,39]. In these studies, bifunctional conjugates of nanoparticles were used. In these conjugates, a part of the nanoparticle surface was occupied by molecules with affinity to the analyte. The other part was occupied by molecules that provided aggregation (for example, streptavidin and biotinylated protein). This approach requires strict control of the conjugate composition to fulfill the efficiency of both processes. In our approach, aggregation-based enhancement of analytical signal in LFIA serodiagnosis is provided by a monofunctional conjugate of a nanoparticle and an immunoglobulin-binding protein. This simplification becomes possible due to the fact that the antibodies are both analytes and triggers of the aggregation process.

In the present study, the approach proposed for signal amplification and improvement of diagnostics was tested to develop a new LFIA format for the detection of antibodies against the RBD antigen of the SARS-CoV-2.

### 3.2. Characterization of the Conjugate of Gold Nanoparticles with Protein A

Protein A of *Staphylococcus aureus* labeled with gold nanoparticles was used as a reagent for the binding of immunoglobulins. To characterize the size and homogeneity of nanoparticles, microphotographs were obtained by TEM. Images of nanoparticles are presented in Figure 2a and the particle size distribution is shown in Figure 2b. The average size of nanoparticles (171 particles were processed) was 31.51 ± 9.27 nm (minimum value—16.78 nm, maximum value—65.16 nm) with an ellipticity of 1.28 ± 0.27. Conjugated nanoparticles were not aggregated in solution, but rather, they were stable during storage and drying on a membrane.

### 3.3. Detection of Specific IgG in Model Solutions

To estimate the process which occurs in the proposed two-stage serodiagnostic LFIA and the possibility of detecting the lower concentration of specific antibodies in blood samples, the knowledge concerning the minimal concentration of detectable antibodies is very important. For this purpose, we detected specific antibodies against the SARS-CoV-2 RBD in model solutions. Humanized rat monoclonal antibodies against the RBD were used as analytes. They contained human Fc fragments to most accurately reproduce the behavior of real immunoglobulins in human biosamples. Figure 3 shows the results of antibody testing by the common LFIA. It was demonstrated that up to 10 ng/mL of specific antibodies can be detected by this method in the absence of non-specific immunoglobulins. The concentration of IgG specific to individual antigens in the blood is typically in the range of 3–50 μg/mL [40,41,42]. Therefore, the test system has a large margin in sensitivity for antibody detection. Additionally, the most likely reason for a possible false-negative test result is signal suppression by non-specific immunoglobulins.

### 3.4. The Influence of Serum Components on the LFIA

To evaluate the effect of biosamples’ components, primarily, non-specific immunoglobulins, on the LFIA results, we used the pooled serum. The pooled serum was diluted with PBST (1:3–1:100) and MAb against RBD (in the same way as the previous section), and then the mixture was added to the model solutions to a final concentration of 10 μg/mL. The mixture was tested using the common and enhanced LFIAs. The obtained results demonstrated that in the case of the common LFIA, the use of undiluted serum or one serum diluted less than 6 times led to the decrease in the signal to an undetectable value (Figure 4a,c (curve I)). Even when the serum was diluted 100 times, the signal was reduced by 8 times relative to the signal registered after the LFIA in the buffer. Following the second stage of the assay, the LFIA signal increased by more than 3 times (Figure 4b,c (curve II)). The comparison of different serum dilutions demonstrates that a 25-fold dilution of serum is sufficient to achieve the maximum signal.

The obtained data are correlated with the regularities described in the previous study [21], which recommends the dilution of serum from 10 to 100 times to minimize the effect of non-specific antibodies on the results of common serodiagnostic LFIA. For further studies, sera diluted 25 times in PBST were used in the analyzed samples.

### 3.5. Comparison of LODs for Two Formats of the LFIA

Using the selected dilution of serum samples, the LODs for specific MAb RBD5313 were estimated for the common and enhanced LFIA. The final concentration of the added antibodies varied from 3 ng/mL to 10 μg/mL. To be specific, when performing the common LFIA, the LOD of specific antibodies in serum was 300 ng/mL, which was 30-fold higher than when detecting antibodies in a buffer solution with the same test system (Figure 5a,c (curve I)). Following the second stage, the LOD decreased by 30 times (Figure 5b,c (curve II)). Therefore, in our case, the proposed enhancement strongly eliminates the effect of non-specific antibodies on the LODs: A 30-fold loss was compensated by a 30-fold improvement. Moreover, the results demonstrated that at lower antibody concentrations in the sample, the enhancement at the second stage of LFIA was increasingly manifested, as compared with high antibody concentrations (values above 1 µg/mL). This fact is particularly important for the detection of the anti-pathogen antibodies in weakly positive samples.

### 3.6. Determination of the Optimal Concentration of the Label Conjugate

As previously noted, blood immunoglobulins are in large quantitative excess relative to the binding capacity of the label conjugate (considering real ratios of these reactants in the test system). Therefore, the higher the concentration of the conjugate, the higher the sensitivity of the test system. However, an increase in the concentration of colloidal particles increases the likelihood of their aggregation, which reduces the availability of immunoglobulin-binding centers. Therefore, the optimal concentration of the conjugate will provide the maximum binding capacity. To obtain this concentration, we performed a series of test strips that differed in the concentration of the pA–GNP conjugate applied prior to the first stage of the assay. Figure 6 shows the testing results of serum samples with a known concentration of specific antibodies by LFIAs differing in concentrations of pA–GNP applied on the test strips (OD_520_ of GNP solutions were 2, 5, 10, and 20). The optimal concentration of the conjugate, which provided the maximum analytical signal and the minimum LOD, corresponded to OD_520_ = 10. This conjugate concentration was used in all our experiments in the future.

### 3.7. Testing of the Developed LFIAs on Blood Samples of Patients with a Confirmed COVID-19 Diagnosis

To validate the developed analytical systems, the positive standard of the National Institute for Biological Standards and Control (Hertfordshire, UK) was first used. As shown in Figure 7, the proposed enhancing approach increased the coloration in the TZ from 6 times to 3 orders of magnitude. The maximum signal was observed in the enhanced LFIA when the serum was diluted from 20 to 40 times.

Following this step, the developed LFIAs were compared with the ELISA in terms of the efficiency of detecting specific antibodies from patients in the samples. For this purpose, a panel of 24 sera was collected, which included 16 sera from patients with confirmed COVID-19 and eight negative sera. This panel has been tested by ELISA.

The resulting panel of positive and negative samples was tested by the LFIA in two formats (see Table 1). Samples providing the color intensities of the TZ above 100 arbitrary units (a.u.), which roughly corresponded to the visual detection threshold, were considered as positive. To be specific, when using the common LFIA, six out of 16 positive samples provided negative results; namely, the diagnostic sensitivity was 62.5%. This value is in accordance with the middle of the sensitivity interval of SARS-CoV-2 LFIAs. Moreover, in accordance with the published reviews, the sensitivity reached levels that vary from 49 to 85% [14,43,44,45]. To be specific, when using the enhanced LFIA, all of the 16 ELISA-positive sera provided positive results (with a diagnostic sensitivity of 100%). Moreover, when testing a panel of negative sera, no positive results were observed (i.e., the diagnostic specificity was 100%). Notably, the presented analytical parameters characterize the assay with an instrumental registration of the results. Visual assessment may lead to inaccurate conclusions in the case of low coloration.

## 4. Conclusions

The present study demonstrates the critical influence of non-specific immunoglobulins on the diagnostic sensitivity of immunochromatographic serodiagnosis and suggests a viable method to eliminate this influence. The proposed format of the serodiagnostic LFIA with two stages of signal generation was tested for the COVID-19 serodiagnosis and showed a signal increase up to three orders of magnitude. At the same time, the 16 positive sera that were tested showed a decrease in the percentage of false-negative results from 37.5% to 0, as compared with the common serodiagnostic LFIA. The implications for the applied two-stage strategy are evident, operator actions are somewhat more complicated and the assay time was extended by 5 min. However, the modified assay still meets the criteria of rapid testing. The applicability of the described approach is not limited to a specific pathogen. Additionally, the proposed solution can be employed to increase the serodiagnostic sensitivity of other diseases.

## Figures and Tables

**Figure 1 biosensors-12-00434-f001:**
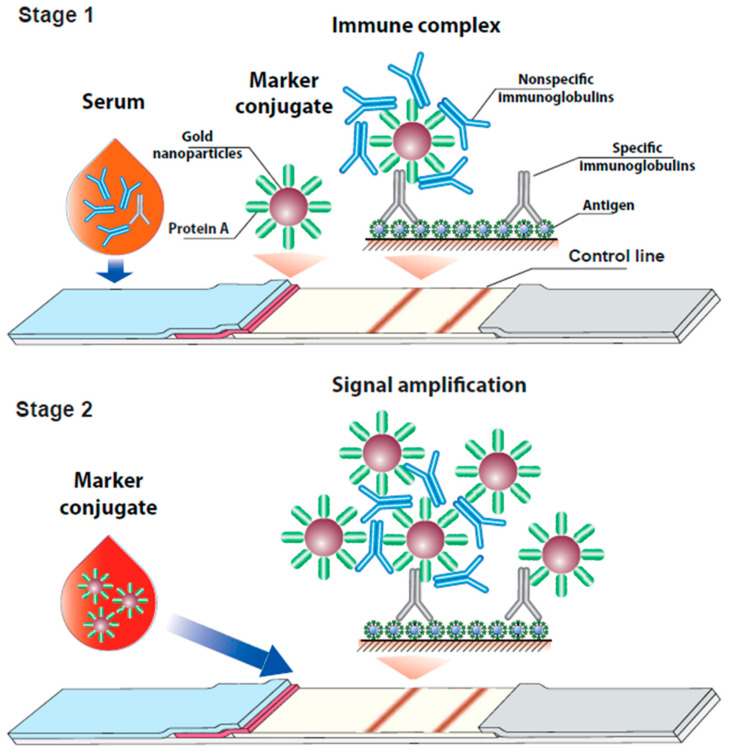
Scheme of the proposed enhanced serodiagnostic LFIA.

**Figure 2 biosensors-12-00434-f002:**
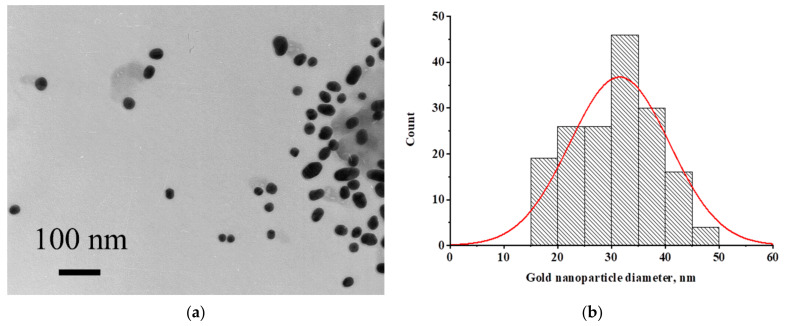
(**a**) Micrograph of the conjugate of gold nanoparticles with staphylococcal protein A. (**b**) Histogram of particle size distribution.

**Figure 3 biosensors-12-00434-f003:**
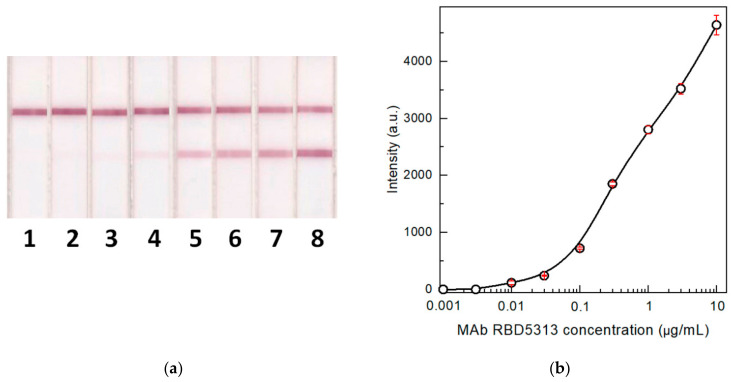
Images of test strips (**a**) after the common LFIA of samples containing 0.003 (1), 0.01 (2), 0.03 (3), 0.1 (4), 0.3 (5), 1 (6), 3 (7), and 10 (8) µg/mL of MAb RBD5313 in PBS and the concentration dependence (**b**) of the assay (*n* = 3). Hereafter, top and bottom colored lines on the images of test strips correspond to the CZ and TZ, respectively.

**Figure 4 biosensors-12-00434-f004:**
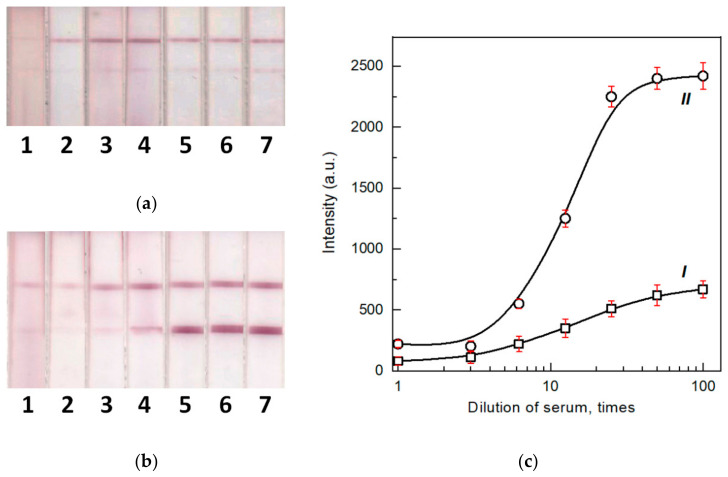
Images of test strips after the common (**a**) and enhanced (**b**) LFIA of samples containing 1- (1), 3- (2), 6.2- (3), 12.5- (4), 25- (5), 50- (6), and 100- (7) -fold dilutions of pooled negative human serum and 10 μg/mL of monoclonal antibodies against RBD. (**c**) Concentration dependence of the common (I) and enhanced (II) LFIA (*n* = 3).

**Figure 5 biosensors-12-00434-f005:**
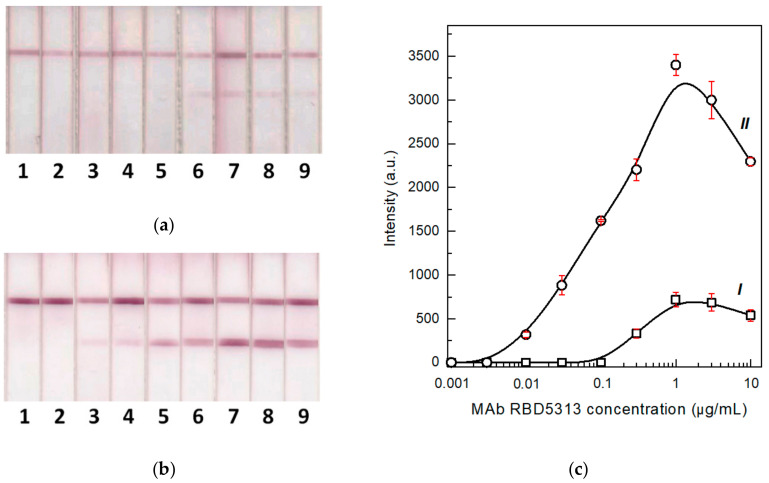
Images of test strips after the common (**a**) and enhanced (**b**) LFIA of samples containing 0 (1), 0.003 (2), 0.01 (3), 0.03 (4), 0.1 (5), 0.3 (6), 1 (7), 3 (8), and 10 (9) µg/mL of MAb RBD5313. (**c**) Concentration dependence of the common (I) and enhanced (II) LFIA (*n* = 3).

**Figure 6 biosensors-12-00434-f006:**
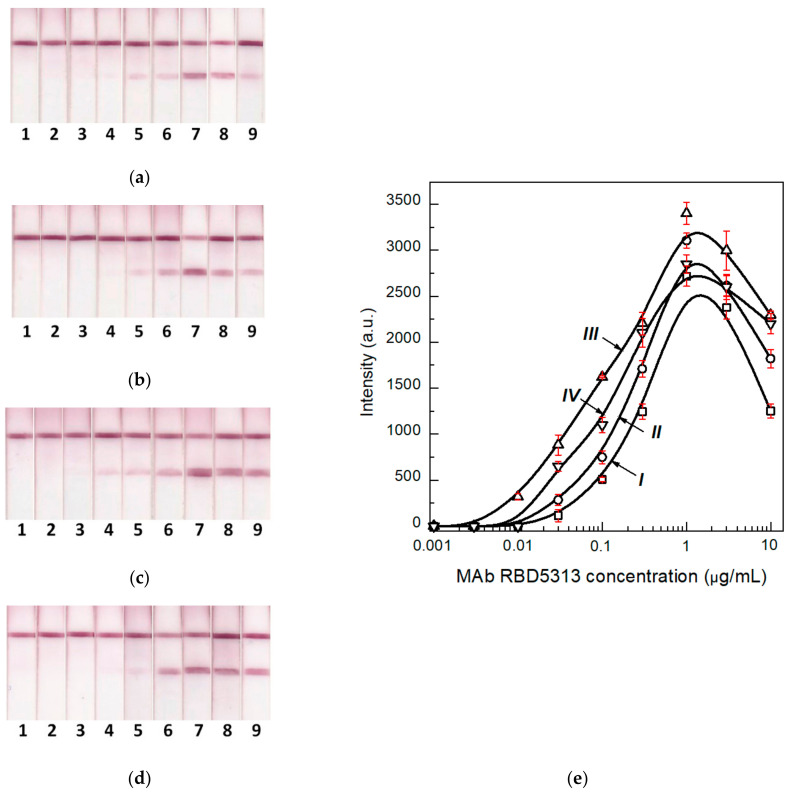
Images of test strips manufactured with the pA–GNP conjugate with OD_520_ = 2 (**a**), 5 (**b**), 10 (**c**), and 20 (**d**) after LFIA of samples containing 0 (1), 0.003 (2), 0.01 (3), 0.03 (4), 0.1 (5), 0.3 (6), 1 (7), 3 (8), and 10 (9) ng/mL of MAb RBD5313. (**e**) Concentration dependence for test systems containing the pA–GNP conjugate with OD_520_ = 2 (I), 5 (II), 10 (III), and 20 (IV) (*n* = 3).

**Figure 7 biosensors-12-00434-f007:**
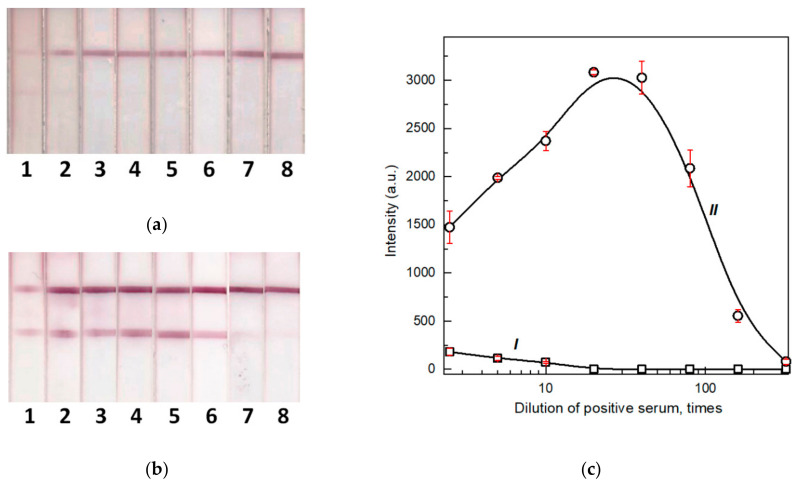
Images of test strips after the common (**a**) and enhanced (**b**) LFIAs of samples of pooled human serum containing antibodies to RBD SARS-CoV-2 diluted by 2.5 (1), 5 (2), 10 (3), 20 (4), 40 (5), 80 (6), 160 (7), and 320 (8) times. (**c**) Concentration dependence of the common (I) and enhanced (II) LFIA (*n* = 3).

**Table 1 biosensors-12-00434-t001:** Comparison of ELISA and LFIA data for 24 human serum samples.

	Positive Sera															
	1	2	3	4	5	6	7	8	9	10	11	12	13	14	15	16
Common LFIA	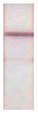	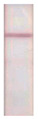	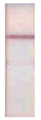	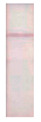	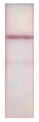	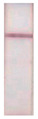	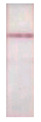	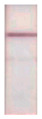	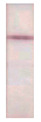	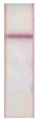	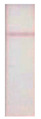	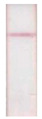	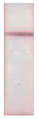	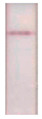	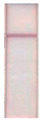	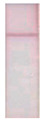
I (a.u.)	208	135	140	135	155	130	142	180	90	85	0	0	0	0	0	35
	+	+	+	+	+	+	+	+	+	-	-	-	-	-	-	-
Enhanced LFIA	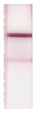	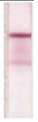	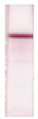	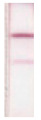	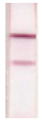	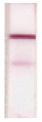	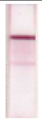	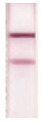	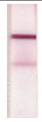	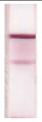	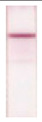	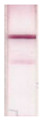	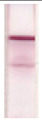	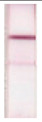	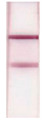	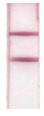
I (a.u.)	585	635	175	525	1650	1285	880	2520	775	1780	125	956	1430	785	2350	3575
	+	+	+	+	+	+	+	+	+	+	+	+	+	+	+	+
ELISA (A 450)	0.807	1.034	0.299	1.004	1.114	0.957	0.747	1.038	0.848	1.124	0.256	0.730	0.932	0.659	0.899	1.072
	**Negative Sera**															
	1		2		3		4		5		6		7		8	
Common LFIA	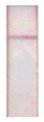		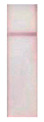		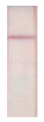		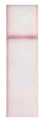		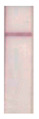		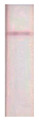		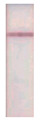		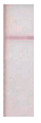	
I (a.u.)	0		0		0		0		0		0		0		0	
	-		-		-		-		-		-		-		-	
Enhanced LFIA	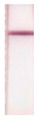		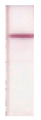		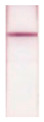		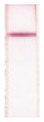		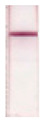		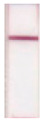		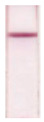		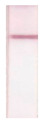	
I (a.u.)	0		65		75		0		55		0		0		0	
	-		-		-		-		-		-		-		-	
ELISA (A 450)	0.125		0.184		0.198		0.093		0.128		0.120		0.191		0.228	

## Data Availability

Data are contained within the article. Initial data of instrumental measurements are available on request from the corresponding author.

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
