# Peer review of "Silent Antibodies Start Talking: Enhanced Lateral Flow Serodiagnosis with Two-Stage Incorporation of Labels into Immune Complexes"

_biosensors, 2022, doi:10.3390/bios12070434_

Round 1

Reviewer 1 Report

The manuscript of Dimitriy V. Sotnikov et al. describes a strategy to improve the analytical performances of lateral flow immunoassays for the detection of pathogen-specifc antibodies.

The manuscript is very interesting and would be of interest for the Biosensors readership.

Below some suggestion for the manuscript improvement.

Introduction.

Page 2, line 51. Please revise “deceases”.

Page 2, lines 68,69. I suggest to mention also the recent https://doi.org/10.1007/s00216-022-03939-2 regarding the use of staphylococcal protein A and streptococcal protein G for the detection of antibodies to SARS‑CoV‑2.

Page 2, lines 76,77. I suggest to mention also the https://doi.org/10.3390/s20226609 in which several combinations have been checked (double antigen sandwich, the use of streptococcal protein G, etc.).

Results and discussion

Page 6, lines 246-248. I am not so sure that the “only” reason for a possible false-negative result is signal suppression by non-specific immunoglobulins. If you agree, I suggest to use something like “the most likely reason”

Page 9, figure 5b. From the image it seems that the test line signal of strip #5 is higher than the one of strip #6.

Table 1. From the results reported in this table it seems that the system is still susceptible to non-specific signal when analyzing negative sera that could lead to false positive results. For example, a very pale signal can be seen also for the negative serum #3 in the common LFIA. Of course, the situation gets worse when employing the enhanced LFIA. The authors “solve” this issue by setting 100 arbitrary units as the Test line threshold value for positivity. Since most of the LFIA for serodiagnosis are based on the visual inspection the authors should discuss this point in a better way or at least underline this phenomenon and specify that instrumental evaluation is needed.

In general.

The authors should specify how they fit the data to obtain the curves.

Are there any disadvantages in applying the two-stage strategy? Please discuss.

For example, is it necessary to have a specific absorbent pad (length, thickness, etc.) in order to avoid liquid backflow?

Author Response

Dear reviewer,

First of all let us thank you for detailed consideration of our manuscript. We have revised it in the accordance with the obtained recommendations. The changed (added, rearranged etc.) fragments are marked in yellow in the revised manuscript. Please find below our comments and cited changes.

Question

Answer / Description of changes

#1. Page 2, line 51. Please revise “deceases”.

The note was accepted (see line 48 of the revised manuscript with the corresponding changes).

#2. Page 2, lines 68,69. I suggest to mention also the recent https://doi.org/10.1007/s00216-022-03939-2 regarding the use of staphylococcal protein A and streptococcal protein G for the detection of antibodies to SARS CoV 2.

The recommended reference has been added (see line 61 of the revised manuscript)

#3. Page 2, lines 76,77. I suggest to mention also the https://doi.org/10.3390/s20226609 in which several combinations have been checked (double antigen sandwich, the use of streptococcal protein G, etc.).

The recommended reference has been added (see line 74 of the revised manuscript)

#4. Page 6, lines 246-248. I am not so sure that the “only” reason for a possible false-negative result is signal suppression by non-specific immunoglobulins. If you agree, I suggest to use something like “the most likely reason”

We agree with the reviewer's remark. The text has been corrected as recommended. (see line 257 of the revised manuscript)

#5. Page 9, figure 5b. From the image it seems that the test line signal of strip #5 is higher than the one of strip #6.

The indicated feature is caused by narrower test line on strip #5 as compared with strip #6. However, we consider total label binding in the test zone that is caused by interaction of all applied reactants independently on variation in their location. Namely, sum of all pixels in the test zone is considered as an analytical signal. This sum is higher for strip #6, as the curve at the right part of Fig. 5 indicates. Moreover, narrow error bars for repeated measurements at the presented curves confirm good reproducibility of the analytical signals considered by us, regardless of variations in the width of the binding zone.

#6. Table 1. From the results reported in this table it seems that the system is still susceptible to non-specific signal when analyzing negative sera that could lead to false positive results. For example, a very pale signal can be seen also for the negative serum #3 in the common LFIA. Of course, the situation gets worse when employing the enhanced LFIA. The authors “solve” this issue by setting 100 arbitrary units as the Test line threshold value for positivity. Since most of the LFIA for serodiagnosis are based on the visual inspection the authors should discuss this point in a better way or at least underline this phenomenon and specify that instrumental evaluation is needed.

Indeed, in some cases very slight background coloration of the test zone after the assay may be observed. Note that the chosen threshold for instrumental detection is close to the threshold of visual identification of binding. The conclusions based on visual observations are may depend on an observer. The instrumental registration overcomes this flaw and therefore it was used in the presented study.

The revised manuscript contains the added explanation of the considered above reasons (lines 349-351):

It should be noted that the presented analytical parameters characterize the assay with instrumental registration of the results. Visual assessment may lead to mistaken conclusions in cases of low coloration.

In general.

#7. The authors should specify how they fit the data to obtain the curves.

The similar recommendation was obtained from another reviewer.

To clarify all sequential processing actions with the obtained initial data, the following text has been added (lines 154-160):

After the LFIA, test strips were scanned using a CanoScan 9000F scanner (Canon, Tochigi, Japan). The obtained digital images were processed by TotalLab TL120 software (Nonlinear Dynamics, Newcastle, UK) to measure the intensity of TZ coloration. The processing of test strips by a TotalLab application included the finding of TZ in the scanned images, generation of color intensity profiles, subtraction of background coloration using the "Background Subtraction" tool, and final registration of the "Volume" value as the analytical signal using the "Band Detection" tool.

#8. Are there any disadvantages in applying the two-stage strategy? Please discuss.

For example, is it necessary to have a specific absorbent pad (length, thickness, etc.) in order to avoid liquid backflow?

The liquid backflow indicated by the reviewer was not observed for our assay protocol using adsorbent membrane AP045 and a sample volume of less than 60 µL. In this case, the liquid remaining in the pores of the working and adsorbing membranes after passing through the main flow of the liquid evaporates noticeably. Therefore, the flow continues to move in the same direction.

The overall estimation of the two-stage assay strategy has been added to the revised manuscript (lines 361-363)

For the used two-stage strategy, operator actions are somewhat more complicated and 5 min are added to the assay time. However, the modified assay still meets the criteria of rapid testing.

Reviewer 2 Report

This manuscript proposed a two-stage LFIA for the serodiagnostic of COVID-19 infection, reducing the effect of nonspecific binding on the results, which presents the enhancement of the coloration intensity by two orders of magnitude compared to the common LFIA format. In gereral, this paper is an interesting and systematic work with comprehensive data supporting. After carefully reading, I think it can be published in Biosensors after major revisions.

 Detailed Comments:

1. Please carefully check and polish the language, there are some usage/grammatical issues.

2. Compared with previous LFIA for the COVID-19 serodiagnosis, how about the sensitivity in this study?

3. How does the visual results translate into mean intensity? Please provide some explanations.

4. Please provide UV-absorption spectra of pure GNP, pA-GNP, and pA-GNP conjugate for comparison.

5. Compared with the signal amplification strategy of the previous studies (Biosensors and Bioelectronics 25 (2010) 1999-2002; Microchim Acta (2017) 184:4189-4195), what is the innovation and advantage of this study.

6. The abstact needs to be condensed and improved.

7. There are some writing errors in the references, the author need to check them totally.

Author Response

Dear reviewer,

First of all let us thank you for detailed consideration of our manuscript. We have revised it in the accordance with the obtained recommendations. The changed (added, rearranged etc.) fragments are marked in yellow in the revised manuscript. Please find below our comments and cited changes.

Question

Answer / Description of changes

#1. Please carefully check and polish the language, there are some usage/grammatical issues.

The final version of the revised manuscript was carefully checked with the involvement of a native English speaking person.

#2. Compared with previous LFIA for the COVID-19 serodiagnosis, how about the sensitivity in this study?

For a comparative assessment of the sensitivity achieved, several recent reviews describing the existing developments of test systems for this disease were taken. From these reviews, general estimates of sensitivity intervals were derived and compared with our development.

Lines 344-346 of the revised manuscript describe the results of the comparison:

That is, the diagnostic sensitivity was 62.5%. This value accords to the middle of the sensitivity interval of SARS-CoV-2 LFIAs. According to published reviews, the reached sensitivities vary from 49 to 85% [14; 43-45].

#3. How does the visual results translate into mean intensity? Please provide some explanations.

The similar recommendation was obtained from another reviewer.

To clarify all sequential processing actions with the obtained initial data, the following text has been added (lines 154-160):

After the LFIA, test strips were scanned using a CanoScan 9000F scanner (Canon, Tochigi, Japan). The obtained digital images were processed by TotalLab TL120 software (Nonlinear Dynamics, Newcastle, UK) to measure the intensity of TZ coloration. The processing of test strips by a TotalLab application included the finding of TZ in the scanned images, generation of color intensity profiles, subtraction of background coloration using the "Background Subtraction" tool, and final registration of the "Volume" value as the analytical signal using the "Band Detection" tool.

#4. Please provide UV-absorption spectra of pure GNP, pA-GNP, and pA-GNP conjugate for comparison.

Conjugate of recombinant staphylococcal protein A with gold nanoparticles was purchased from Arista Biologicals (Allentown, USA). Thus we have not data about initial reactants that were used by the manufacturer and cannot provide the requested comparison.

#5. Compared with the signal amplification strategy of the previous studies (Biosensors and Bioelectronics 25 (2010) 1999-2002; Microchim Acta (2017) 184:4189-4195), what is the innovation and advantage of this study.

The following text with the discussion of the indicated papers has been added to the revised manuscript (lines 219-228):

Aggregation of functionalized nanoparticles was earlier used in LFIA to enhance the analytical signal [38,39]. In these studies, bifunctional conjugates of nanoparticles were used. In such conjugates, a part of the nanoparticle surface was occupied by molecules having affinity to the analyte. The other part was occupied by molecules that provided aggregation (for example, streptavidin and biotinylated protein). This approach requires strict control of the conjugate composition to fulfill the efficiency of the both processes. In our approach, aggregation-based enhancement of analytical signal in LFIA serodiagnosis is provided by a monofunctional conjugate of a nanoparticle and an immunoglobulin-binding protein. This simplification becomes possible because antibodies are both analytes and triggers of the aggregation process.

#6. The abstract needs to be condensed and improved.

The Abstract has been reduced and redesigned as recommended.

#7. There are some writing errors in the references, the author need to check them totally.

Formats of the references and the data presented in them were carefully checked

Round 2

Reviewer 1 Report

The authors have addressed all the issues raised by the reviewers improving their manuscript. Theregore, I suggest the manuscript publication.

Reviewer 2 Report

The manuscript could be published at the current form.